# The Actin-Family Protein Arp4 Is a Novel Suppressor for the Formation and Functions of Nuclear F-Actin

**DOI:** 10.3390/cells9030758

**Published:** 2020-03-19

**Authors:** Shota Yamazaki, Christian Gerhold, Koji Yamamoto, Yuya Ueno, Robert Grosse, Kei Miyamoto, Masahiko Harata

**Affiliations:** 1Laboratory of Molecular Biology, Graduate School of Agricultural Science, Tohoku University, Sendai 980-0845, Japan; 2Terahertz Sensing and Imaging Research Team, RIKEN Center for Advanced Photonics, Sendai 980-0845, Japan; 3Friedrich-Miescher Institute for Biomedical Research, Maulbeerstr. 66, 4058 Basel, Switzerland; 4Institute of Experimental and Clinical Pharmakology and Toxicology, Faculty of Medicine, University of Freiburg, Alberstrasse 25, 79104 Freiburg, Germany; 5Laboratory of Molecular Developmental Biology, Faculty of Biology-Oriented Science and Technology, Kindai University, Wakayama 649-6493, Japan

**Keywords:** nuclear architecture, nuclear actin, actin-related protein, nucleoskeleton, epigenetics

## Abstract

The crosstalk between actin and actin-related proteins (Arps), namely Arp2 and Arp3, plays a central role in facilitating actin polymerization in the cytoplasm and also in the nucleus. Nuclear F-actin is required for transcriptional regulation, double-strand break repair, and nuclear organization. The formation of nuclear F-actin is highly dynamic, suggesting the involvement of positive and negative regulators for nuclear actin polymerization. While actin assembly factors for nuclear F-actin have been recently described, information about inhibitory factors is still limited. The actin-related protein Arp4 which is predominantly localized in the nucleus, has been previously identified as an integral subunit of multiple chromatin modulation complexes, where it forms a heterodimer with monomeric actin. Therefore, we tested whether Arp4 functions as a suppressor of nuclear F-actin formation. The knockdown of Arp4 (Arp4 KD) led to an increase in nuclear F-actin formation in NIH3T3 cells, and purified Arp4 potently inhibited F-actin formation in mouse nuclei transplanted into *Xenopus laevis* oocytes. Consistently, Arp4 KD facilitated F-actin-inducible gene expression (e.g., *OCT4*) and DNA damage repair. Our results suggest that Arp4 has a critical role in the formation and functions of nuclear F-actin.

## 1. Introduction

Actin has structural and regulatory roles in the cytoplasm and the nucleus. The nuclear import of actin is regulated by importin 9 [1], and exportin 6 exports actin to the cytoplasm [2]. Nuclear G-actin is included in multiple chromatin remodeling and histone modification complexes, and is involved in the regulation of genome functions [3]. Nuclear G-actin also regulates the serum response factor (SRF) pathway through MAL (MRTF-A), a co-activator of SRF [4,5]. MAL bound with nuclear G-actin is exported to the cytoplasm, and as a result, expression of the target genes of SRF is repressed [5].

Nuclear F-actin also plays various roles in the nucleus, including in transcriptional regulation [6]. For example, the formation of nuclear F-actin decreases the nuclear G-actin pool, causing activation of SRF target genes through the retention of MAL in the nucleus [4]. More specifically, nuclear F-actin is formed during the process of gene reprogramming of somatic cell nuclei transferred into oocytes, and F-actin formation is required for transcriptional reprogramming through the reactivation of the pluripotency gene *OCT4* [7]. Moreover, by associating with β-catenin, nuclear F-actin increases the amount of β-catenin in the nucleus. This enhances the binding of β-catenin to chromatin and activates Wnt/β-catenin target genes [8]. In addition to these transcriptional regulations, dynamic nuclear F-actin is formed under various conditions, for example, in response to DNA damage [9,10,11], upon cell spreading [12], calcium-triggered chromatin dynamics and at mitotic exit [13,14].

Although the cell nucleus contains a sufficient amount of the actin needed for polymerization [15], nuclear actin polymerization depends on specific signals and appears to be tightly regulated. Thus, a dynamic transition between the G- and F-actins appears to exist in the nucleus [16]. F-actin promoting factors likely compete with potent suppressors of nuclear F-actin in controlling nuclear actin dynamics. Various F-actin promoters, including N-WASP, WAVE1, formin, and profilin have been reported [4,7,10,17,18,19]. Nuclear proteins, including lamin, emerin, LINC complex, and the nuclear pore complex also promote nuclear F-actin formation [12,18,20]. Although cofilin was recently shown to play an important role in the disassembly of nuclear F-actin at mitotic exit [13], information about inhibitory factors of nuclear F-actin remains limited.

Crosstalks among the actin family members also provide an important platform for the transition of G- and F-actins. The actin family consists of conventional actin isoforms and structurally similar actin-related proteins (Arps) [21,22,23,24]. These Arps are classified into 10 subfamilies, Arp1 to Arp10, and among them, Arp2 and Arp3 form the core of the Arp2/3 complex which binds to actin and facilitates actin polymerization by serving as an actin nucleation site [25]. It was recently reported that, also in the nucleus, the Arp2/3 complex facilitates F-actin formation [10,17,26].

Among the 10 Arp subfamilies, Arp4–Arp9 are found predominantly in the nucleus [3]. These nuclear Arps reside in multiple chromatin remodeling and histone modification complexes, in most cases, together with actin, suggesting that actin regulation by nuclear Arps occurs in the nucleus. Arp4 (also known as ACTL6 or BAF53), an actin family protein that is predominantly localized in the nucleus, is an evolutionarily conserved protein [22,27,28]. Arp4 is an essential component of chromatin remodeling and histone modification complexes, including SWI/SNF, RSC, INO80, SWR1, and NuA4 complexes in budding yeast, and SWI/SNF, INO80, SRCAP, and Tip60 complexes in humans [3,29,30]. Interestingly, Arp4 forms a heterodimer with actin in these chromatin modulating complexes [31]. A previous study showed that recombinant yeast Arp4 inhibited actin polymerization in vitro by forming a complex with G-actin [22]. In addition, human Arp4 was shown to bind directly to G-actin [28].

In the present study, we show that Arp4 is a novel suppressor of nuclear F-actin formation in mammalian cells. Our finding implies a novel role for Arp4 in regulating the dynamics and function of nuclear actin.

## 2. Materials and Methods

### 2.1. Cell Lines and Plasmid Constructs

NIH3T3 cells were cultured in Dulbecco’s modified Eagle’s medium (DMEM) (Wako, Osaka, Japan) supplemented with 10% fetal bovine serum and antibiotics (penicillin and streptomycin) at 37 °C in a 5% CO_2_ humidified atmosphere. To silence Arp4, the following two 21-residue long nucleotide sequences, corresponding to Arp4 cDNA nucleotides 293–313 and 448–468, respectively, were expressed individually as siRNAs in NIH3T3 cells using the BLOCK-iT RNAi system (Invitrogen, Carlsbad, CA, USA): siArp4-1 (293–313), 5’-CTTTCTACCACCATACCAATA-3’; and siArp4-2 (448–468), 5’-AACTATCCCAGTCTTCAACCA-3’. Plasmid pcDNATM6.2-GW/EmGFP-siR-neg-control, which contains an insert that can form a hairpin structure that is processed into a mature siRNA but is predicted not to target any known vertebrate gene, was used to create control cells. An ectopic Arp4 plasmid with a mutated targeting site (5’-CTCTCAACGACCATTCCGATA-3’) for siArp4-1 was used for transfection with siArp4-1 to probe potential off-target effects of siRNA knockdown. For visualization of nuclear F-actin, an anti-actin-chromobody tagged with a nuclear localization signal and mCherry (nAC-mCherry) was used. Plasmid constructs were transfected into NIH3T3 cells by electroporator, as described below.

### 2.2. Introduction of Constructs into Cells and Fluorescence Microscopy

NIH3T3 cells grown on a 6 cm dish similar to subconfluency were used for the transfection of plasmid constructs. The growth of cells was determined by the trypan blue exclusion method, using a TC-20 automated cell counter (Bio-Rad, Hercules, CA, USA), and 1 × 10^5^ cells were used for electroporation. Ten µg of plasmid was transfected into cells by electroporation with a NEPA21 electroporator (NEPAGENE, Chiba, Japan). Poring pulse voltage = 175 V, pulse length = 3 ms, pulse interval = 50 ms, number of pulses = 2, and decay rate = 10%. Transfer pulse voltage = 20 V, pulse length = 50 ms, pulse interval = 50 ms, number of pulses = 5, and decay rate = 40%. For observation of nuclear F-actin by fluorescence microscopy, 10 µg of nAC-mCherry and 10 µg of control or Arp4 siRNA were used to co-transfect 1 × 10^5^ cells by electroporation. The cells were grown on a glass coverslip for 24 h, which was placed inside the wells of a 6-well culture plate (BM Equipment, Tokyo, Japan). The cells were fixed with 4% paraformaldehyde/phosphate-buffered saline (PBS) and, then, permeabilized with 0.5% Triton X-100 in PBS for 10 min at room temperature. Nuclei were stained with 4′, 6-diamidino-2-phenylindole (DAPI) (Invitrogen, Carlsbad, CA, USA). Images were collected and analyzed on an IX-83 wide-field microscope (Olympus) equipped with a 100× lens (UPlanSApo, numerical aperture of 1.40, Olympus, Tokyo, Japan). For control and Arp4-KD cells, images were captured using identical exposure times. Signal intensities of nAC-mCherry and DAPI were measured by Fiji image analysis software.

### 2.3. Antibodies and Immunofluorescence Staining

Cells were transfected with the indicated constructs by electroporation and cultured on a glass coverslip at 37 °C in 5% CO_2_ humidified atmosphere for 24 h. For immunofluorescence staining, incubated cells were fixed with 4% paraformaldehyde/phosphate-buffered saline (PBS) and then permeabilized with 0.5% Triton X-100 in PBS for 10 min at room temperature. For the antibody reaction, cells were treated with an anti-rabbit β-catenin (ab16051, Abcam, Cambridge, UK) or an anti-phosphoH2A.X (Ser139) (Millipore, Billerica, MA, USA) in 4% bovine serum albumin/PBS for 1 h at room temperature or overnight at 4 °C. Then, cells were treated with anti-rabbit Alexa 405 (ab175651, Abcam, Cambridge, UK) or Alexa Fluor 594 mouse IgG (Molecular Probes, Eugene, OR, USA) for 1 h at room temperature. For each cell line, images were captured using identical exposure times. Signal intensities of β-catenin, γH2A.X, and DAPI were measured by Fiji image analysis software (U.S. National Institutes of Health, Bethesda, MD, USA).

### 2.4. Quantitative RT-PCR Analysis

Total RNA was reverse-transcribed to cDNA using the Super Script III First-Strand Synthesis System (Invitrogen, Carlsbad, CA, USA). Real-time PCR was carried out using SYBR-Green PCR Master Mix (Thermo Fisher Scientific, Waltham, MA, USA) in a total volume of 10 μL (for cultured cells) or 15 μL (for oocyte reprogramming assays). For cultured cells, values were normalized to that of the endogenous control, glyceraldehyde-3-phosphate dehydrogenase (GAPDH), and compared to control samples using the ΔΔCt method. For the *Xenopus* transcriptional reprogramming assay, quantification was performed using embryonic stem cell cDNA within the log-linear phase of the amplification curve. A relative transcript level for every gene was measured and compared. The primer sequences used for this analysis are summarized in Appendix A.

### 2.5. Dual-Luciferase Reporter Assay

This assay was performed using the Dual-Luciferase Reporter Assay System (Promega, Madison, WI, USA), following the manufacturer’s instructions. Briefly, a TCF/LEF reporter construct or an SRF reporter construct was introduced into cells together with a construct for EmGFP-siR-neg-control or with siArp4-1 using FuGENE HD (Boehringer). As an internal control for transfection efficiency, a pRL-TK renilla luciferase construct was used for co-transfection in each case. All experiments were performed at least three times with independent cell cultures.

### 2.6. Chromatin Immunoprecipitation (ChIP) Assay

A ChIP assay was performed according to the protocol supplied with the ChIP reagents (Nippon Gene, Tokyo, Japan). Briefly, NIH3T3cells transfected either with EmGFP-siR-neg-control or with siArp4-1 were crosslinked using 1% formaldehyde and were used to isolate nuclear extracts. The β-catenin-bound chromatin fraction in the sonicated nuclear lysates was immunoprecipitated with anti-β-catenin antibody (AF1329, R&D Systems, Minneapolis, MN, USA) bound to Dynabeads Protein G (Invitrogen, Carlsbad, CA, USA). Purified DNA from the immunoprecipitated fraction was subjected to real-time PCR using the primers listed in Appendix A.

### 2.7. Nuclear Transfer and Injection of Oocytes

C2C12 cells were permeabilized with streptolysin O (Sigma-Aldrich, St. Louis, MO, USA) and used as donor cells for nuclear transfer (NT). Approximately 300 permeabilized cells in yArp4-mCherry-containing buffer or control buffer were injected into the germinal vesicle of *Xenopus laevis* oocytes [7,19]. Before NT, oocytes were loaded with overexpressed proteins by injecting GFP-UtrCH (40.8 ng/40.8 nL) and histone H2B-CFP (4.6 ng/4.6 nL) mRNAs. NT oocytes were incubated at 18 °C for 48 h before confocal microscopy and RT-qPCR. Germinal vesicles (GVs) of NT oocytes were dissected in mineral oil for observation under a confocal microscope. Confocal analysis was carried out on a Zeiss 510 META confocal LSM microscope (Carl Zeiss, Oberkochen, Germany) equipped with argon (488 nm), HeNe (543 nm), and MaiTai (740 nm) lasers. For RT-qPCR analyses, 6 oocytes were pooled and RNA was purified, as described previously in [7].

### 2.8. Statistical Analyses

In transcriptional assays, technical replicates were performed on all samples and an average of the duplicate was regarded as the value of one sample. The number of biological replicates is shown as *n*. In the box plot, the number of cells is shown as *n*. Statistical significances were calculated by F- and T-test for the transcriptional assay, nuclear F-actin formation, and box plot.

## 3. Results and Discussion

### 3.1. Depletion of Arp4 Induces Aberrant Endogenous Actin Structures in the Nucleus

To test whether nuclear F-actin formation is regulated by a nuclear Arp, Arp4, we knocked down Arp4 by RNA interference (Appendix A) and observed nuclear F-actin using an anti-actin-chromobody tagged with a nuclear localization signal (nAC) [12]. In control cells, thin nuclear actin filaments were observed in a small portion of the population (Figure 1A,B). We observed that nuclear F-actin bundles were thickened by Arp4 KD (Figure 1A). Indeed, the intensity of nuclear F-actin was increased by Arp4 KD in a statistically significant manner (Appendix A). In addition, Arp4 KD increased the number of cells possessing nuclear F-actin (Figure 1B). These results suggest that Arp4 negatively affects the formation of nuclear F-actin. A FACS analysis revealed that the proportion of cells in each cell cycle stage is not changed by Arp4 KD (Appendix A), and therefore Arp4 KD did not affect the formation of nuclear F-actin indirectly through the change of cell cycle progression.

Since Arp4 is included in multiple chromatin remodeling complexes, we examined whether Arp4 KD would affect the expression of actin and the actin transporters importin 9 and exportin 6. Quantitative RT-PCR analysis showed no changes in the expression levels of actin, importin 9, and exportin 6 genes in the Arp4 KD cells (Figure 1C). As Arp4 is known to directly bind to actin and impede actin polymerization in vitro [22], the above results support the idea that the interaction between Arp4 and actin inhibits the formation of F-actin in the nucleus.

### 3.2. Arp4 Suppresses Nuclear F-Actin Formation in Xenopus Oocytes

*Xenopus* oocytes contain large amounts of nuclear actin. As shown previously, nuclear F-actin is formed in the process of gene reprogramming of somatic cell nuclei transferred into the *Xenopus* oocyte, and F-actin formation is an essential step of gene reprogramming of *Oct4* [7]. We used this system to further analyze the role of Arp4 in suppressing nuclear F-actin formation and its role in regulating gene expression through actin dynamics. We induced gene reprogramming by transplanting mouse C2C12 myoblast cell nuclei into *Xenopus* oocytes at the GV stage. This NT triggers the initiation of gene reprogramming without cell division and de novo protein synthesis [32,33]. To analyze the effect of Arp4 on nuclear F-actin formation in the course of gene reprogramming, we injected purified yeast Arp4 into the nuclei-transplanted *Xenopus* oocytes and observed its effect on nuclear F-actin formation. The F-actin probe GFP-UtrCH and histone H2B-CFP mRNA were injected into the recipient *Xenopus* oocytes prior to the injection of Arp4-mCherry and C2C12 nuclei (Figure 2A). Nuclear F-actin was observed in the transplanted nuclei after the injection of the control solution. The injection of a small amount of Arp4 substantially decreased the level of F-actin in the nuclei, and more importantly, increased the amount of injected Arp4, resulting in the disappearance of the nuclear F-actin (Figure 2B).

The formation of F-actin in the nuclei transferred into *Xenopus* oocytes is required for the activation of *Oct4*, but not for other genes, including *Sox2* and *Utf1* [7]. The injection of Arp4 did indeed inhibit the expression of the *Oct4* gene. In contrast, expression levels of other genes, as well as that of the control *Gapdh* gene, were not significantly changed (Figure 2C). These results further suggest an inhibitory role for Arp4 in nuclear F-actin formation, and also suggest its role in transcription regulation via nuclear actin dynamics.

### 3.3. Effects of Arp4 Knockdown on F-Actin-Inducible Transcription

The formation of nuclear F-actin induces *OCT4* expression in cultured human cells [8,34]. Consistent with this observation and the results shown in Figure 2C, knockdown of Arp4 induced expression of the *OCT4* gene, but not of the *SOX2*, *UTF1*, and *GAPDH* genes (Figure 3A). The increase of nuclear F-actin, by exogenously expressing NLS-tagged actin [34], further enhanced the expression of the *OCT4* gene in the Arp4 KD cells (Figure 3B). These results suggest that Arp4 antagonizes *OCT4* gene expression by inhibiting nuclear actin polymerization.

It has been shown that nuclear F-actin increases nuclear SRF activity [4,35]. Therefore, next, we monitored the activity of SRF in Arp4 KD cells. A luciferase assay using a reporter plasmid containing an SRF-responsive CArG element showed increased SRF activity upon Arp4 KD (Figure 3C). Expression of the genomic SRF targets *IGF1* and *ACTA2* was also increased in Arp4 KD cells (Figure 3D). These results further support the idea that Arp4 depletion affects nuclear actin dynamics and hence actin-dependent transcription.

### 3.4. Depletion of Arp4 Activates Wnt/β-Catenin-Targeting Genes

Nuclear F-actin formed by expressing NLS-actin tethers β-catenin in the nucleus and activates β-catenin-targeting promoters [8]. To test whether the F-actin visualized with nAC-mCherry in Arp4 KD cells exhibits the same property, we analyzed the localization and activity of β-catenin in the nucleus from Arp4 KD cells. As shown in Figure 4A, nuclear β-catenin was colocalized with F-actin bundles induced by Arp4 KD. Notably, knockdown of Arp4 also induced the appearance of β-catenin in the nucleus (Figure 4B). Consistent with this notion, the expression of a reporter plasmid containing a TCF/LEF element showed increased β-catenin activity in Arp4 KD cells (Figure 4C). Consequently, the expression levels of the genomic targets of β-catenin, namely *OCT4*, *CCND1*, *AXIN2*, and *TCF7* genes, were augmented in Arp4 KD cells (Figure 3B and Figure 4D). Furthermore, consistent with the previous observation that nuclear F-actin increased the binding of β-catenin to the promoters of *OCT4* and *CCND1* [8], the increased binding of β-catenin to these genes in Arp4 KD cells was observed (Figure 4E). These results suggest that Arp4 has a role in gene expression through suppressing the formation of nuclear F-actin.

### 3.5. Arp4 Knockdown Affects DNA Double-Strand Break Repair

Nuclear F-actin is required for DNA double-strand break (DSB) repair [17]. Hence, we tested DSB repair by counting γH2A.X foci after DNA damage induction with zeocin, with or without Arp4 KD. While Arp4 KD did not affect γH2A.X foci formation before the treatment with zeocin, Arp4 KD significantly decreased γH2A.X foci induced by zeocin (Figure 5). The knockdown of Arp3, which facilitates F-actin formation, caused an opposite effect on γH2A.X foci [10]. These results suggest that nuclear F-actin increased by Arp4 KD can function during DSB repair.

### 3.6. Arp4 is Involved in Genome Functions as a Novel Suppressor of F-Actin Formation

In this study, we identified Arp4 as a member of the actin family of proteins that suppresses F-actin formation in the nucleus. Two other actin family members, Arp2 and Arp3, are known to facilitate actin polymerization in the cytoplasm and also in the nucleus [10,11]. The basal structures of Arps2/3 and -4 are highly similar to actin, and both bind to actin [6,24]. However, the former binds to the pointed end while the latter binds to the barbed end of an actin molecule [22]. Thus, the different characteristics of these Arps would lead to their antagonizing functions for actin polymerization. This crosstalk among actin-family members could provide an efficient mode of regulation of nuclear actin dynamics, as well as F-actin structures. While the Arps2/3 complex is dominant in the cytoplasm, Arp4 is accumulated in the nucleus [36]. Thus, the accumulation of Arp4 could contribute to the dynamic nature and appearance of nuclear F-actin.

In addition to suppressing nuclear F-actin formation, Arp4 functions as an essential component of chromatin remodeling and histone acetylation complexes together with G-actin [3,30,37,38]. Arp4 has the histone binding activity [39] and its involvement is also suggested in the assembly and disassembly of these complexes, depending on its ATP- or ADP-binding state [40,41]. Therefore, Arp4 acts not only as a suppressor of F-actin, but also as a multifunctional regulator of chromatin modulating complexes. Arp4 could balance functions of F-actin and G-actin in nuclear organization and chromatin modulations, respectively.

Previous studies have shown that the expression of Arp4 changes during cell differentiation, and that depletion of Arp4 affects the progression of cell differentiation [42,43,44]. Furthermore, human Arp4 (ACTL6A) is overexpressed in various types of cancer cells, and regulates tumor growth and progression [45,46,47,48]. However, the biologic roles of Arp4 in carcinogenesis remain unknown. It has been shown that dysregulation of the level and structures of nuclear actin contributes to diseases such as cancer [49,50]. Interestingly, nuclear F-actin binds to normal p53, but not to its cancer-associated mutants [51]. Until now, Arp4 is supposed to contribute to carcinogenesis through its essential roles in chromatin remodeling and histone modification complexes. However, together with our observations supporting a nuclear role for Arp4 in regulating the dynamics of nuclear F-actin, it is suggested that the overexpression of Arp4 contributes to carcinogenesis through its effect on nuclear F-actin formation and also on nuclear organization [52,53].

## 4. Conclusions

In this study, we showed that the predominant nuclear Arp, Arp4, regulates actin dynamics by suppressing the formation of nuclear F-actin. Our results implicate Arp4 in regulating genome functions through inhibitory regulations on nuclear F-actin assembly.

## Figures and Tables

**Figure 1 cells-09-00758-f001:**
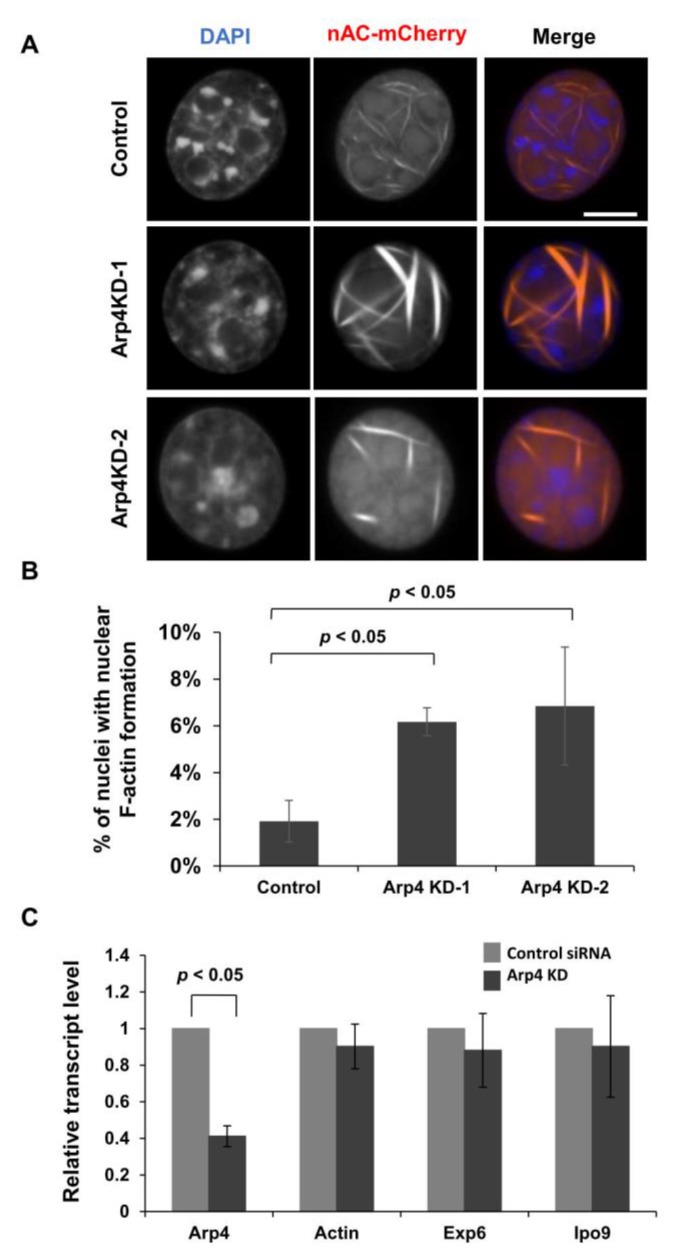
Knockdown of Arp4 increased nuclear F-actin. (**A**) Confocal microscopy images of NIH3T3 cells transfected with control siRNA (control), Arp4-1 siRNA (Arp4 KD-1), or Arp4-2 siRNA (Arp4 KD-2). Nuclear F-actin formation was detected by expressing the nuclear actin probe nAC-mCherry. Nuclei were labeled using DAPI (blue). Bar = 10 μm; (**B**) Quantifications of nuclear F-actin formation in control cells (control+nAC-mCherry) and Arp4 KD cells (Arp4 KD-1 or Arp4 KD-2+nAC-mCherry) cells. For the quantification, 309 control cells, 308 Arp4 KD-1 cells, and 306 Arp4 KD-2 cells were analyzed. Data shown are mean ± S.D. (*n* ≥ 3); (**C**) Quantitative RT-PCR analysis of Arp4, actin, exportin 6 (Exp6), and importin 9 (Ipo9) mRNAs in control and Arp4 KD-1 cells. Results shown are values relative to those in the control cells.

**Figure 2 cells-09-00758-f002:**
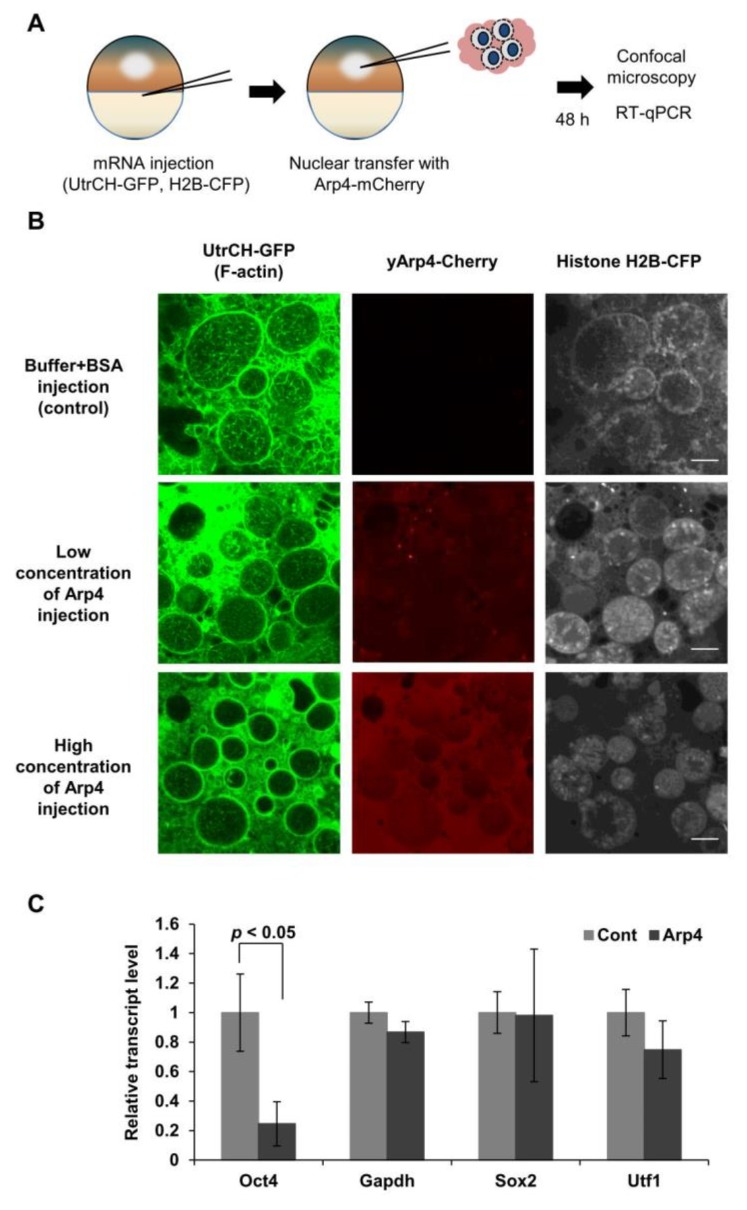
Injection of purified Arp4 suppresses F-actin formation and *Oct4* expression in mouse nuclei transplanted into *Xenopus* oocytes. (**A**) A schematic diagram of the experimental strategy used to visualize the nuclear F-actin formed in the transplanted nuclei. Dissected GVs were alive in mineral oil and were sufficiently transparent to permit observation of interior structures by confocal microscopy; (**B**) The polymerization of actin in the transplanted nuclei was suppressed by injected Arp4-mCherry in a concentration-dependent manner; (**C**) Levels of mouse *Oct4*, *Gapdh*, *Sox2*, and *Utf1* mRNAs were quantified in the transplanted nuclei after injecting Arp4-mCherry into oocytes. The mRNA levels (mean ± SEM, *n* = 3) shown are relative to those in the control nuclei.

**Figure 3 cells-09-00758-f003:**
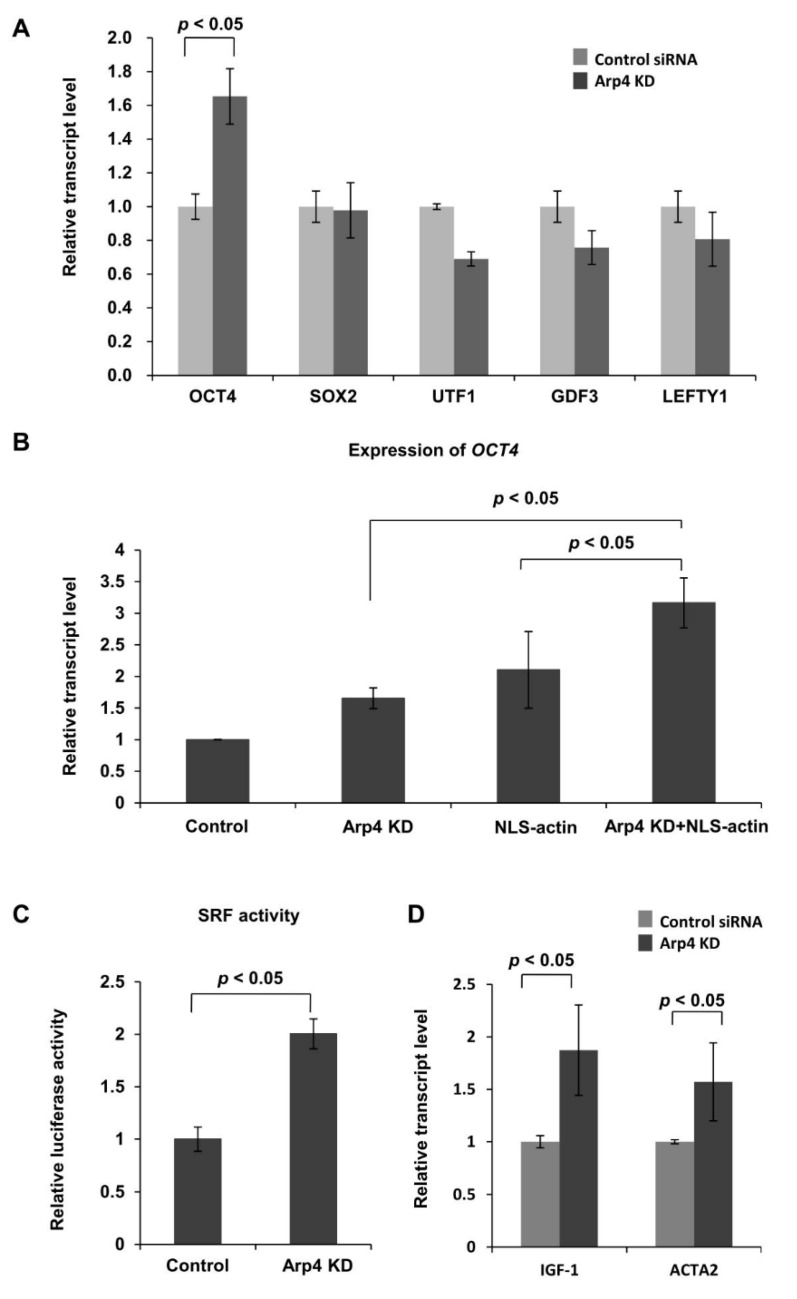
Effects of Arp4 KD on gene expression. (**A**) Expression levels of *OCT4*, *SOX2*, *UTF1*, *GDF3*, and *LEFTY1* mRNAs in Arp4 KD cells (Arp4-1 siRNA) were quantified and are shown relative to their respective levels in the control cells expressing control siRNA; (**B**) The expression level of *OCT4* in control non-treated NIH3T3 cells, Arp4 KD cells, cells ectopically expressing NLS-actin (NLS-actin), and Arp4 KD cells ectopically expressing NLS-actin (Arp4 KD+NLS-actin). Values shown are relative to their respective levels in control cells; (**C**) The SRF-RE reporter plasmid was introduced into NIH3T3 cells expressing control-siRNA (control) or Arp4-1 siRNAs (Arp4 KD), and SRF responsive luciferase activity in each sample was measured as described in the Materials and Methods section. The plot shows activity relative to that in the control cells, the value for which was assigned as 1.0; (**D**) Quantitative RT-PCR analysis of mRNAs of SRF-targeting genes *IGF-1* and *ACTA2* in control and Arp4 KD cells. The expression level of each gene was normalized with respect to that of the *GAPDH* gene. Data shown are mean ± S.D. (*n* ≥ 3).

**Figure 4 cells-09-00758-f004:**
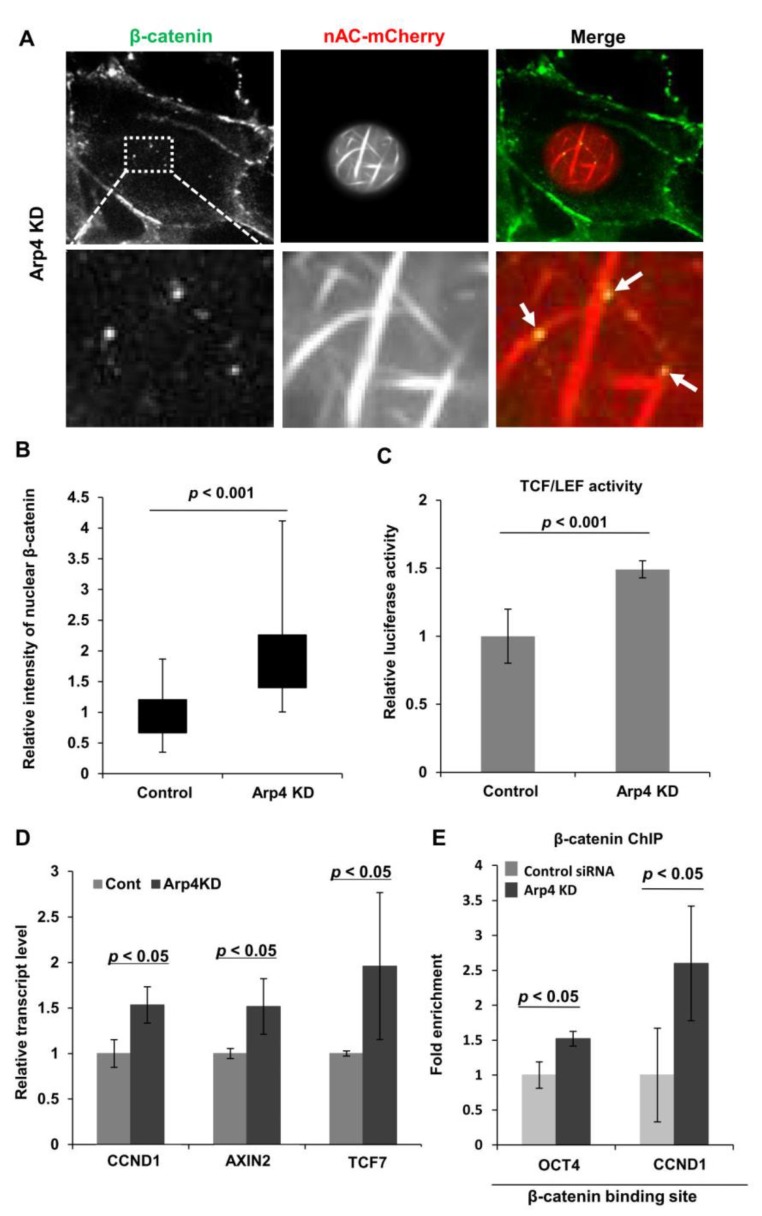
Arp4 is involved in β-catenin-dependent gene expression through suppressing nuclear F-actin formation. (**A**) NIH3T3 cells expressing siArp4-1 (Arp4 KD) and nAC-mCherry were immunostained with a specific β-catenin antibody. The arrows show colocalization of nuclear F-actin and β-catenin. Bar = 5 μm; (**B**) Signal intensities of β-catenin in the nucleus and cytoplasm were measured in control or Arp4 KD cells, and the relative amount of nuclear β-catenin was shown. Box plot, the amount of β-catenin in the nucleus relative to that in the cytoplasm and *n* = 52; (**C**) A TCF/LEF reporter plasmid construct was introduced into the control or Arp4 KD cells, and TCF/LEF responsive luciferase activity was measured in each sample. Luciferase activity shown is relative to that in the control cells; (**D**) Quantitative RT-PCR analysis of mRNAs of TCF/LEF targeting genes, *CCND1*, *AXIN2*, and *TCF7*. The expression level of each gene in control siRNA and Arp4 KD cells was normalized with respect to that of the *GAPDH* gene; (**E**) The binding of endogenous β-catenin to its targeting sites in *OCT4* and *CCND1* genes was analyzed by quantitative ChIP assay, using an antibody specific for β-catenin. The values shown are relative to those in control cells. Data shown are mean ± SD (*n* ≥ 3).

**Figure 5 cells-09-00758-f005:**
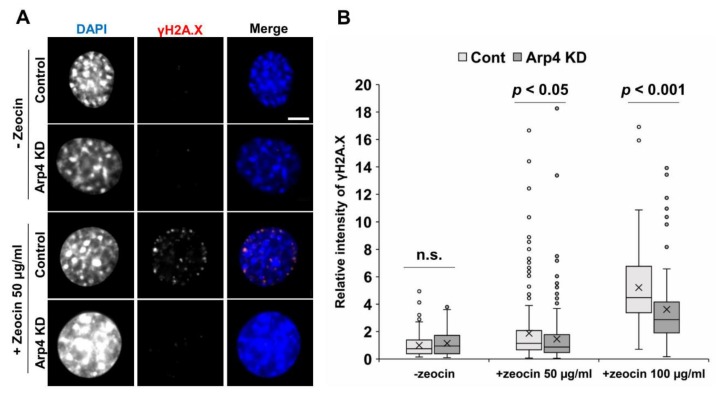
The effect of Arp4-KD in DNA double-strand break repair. Control siRNA (control or cont) or siArp4-1 (Arp4 KD) was introduced into NIH3T3 cells, and cells were incubated for 24 h. The cells were, then, treated with zeocin (50 or 100 µg/mL) for 1 h and subjected to γH2A.X staining a specific antibody and DAPI staining. The cells were observed under a fluorescent microscope (**A**), and the relative fluorescence intensity of γH2A.X to that of DAPI was determined using Fiji image analysis software (**B**). Box plot, the signal intensity of γH2A.X relative to that of DAPI and *n* > 88.

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
