# Peer review of "The Actin-Family Protein Arp4 Is a Novel Suppressor for the Formation and Functions of Nuclear F-Actin"

_cells, 2020, doi:10.3390/cells9030758_

Round 1
Reviewer 1 Report
The study by Yamazaki et al., shows that in mammalian cells, the actin related protein Arp4 (also known as Baf53) inhibits actin polymerization and suppresses the formation of F-actin. This is a novel finding and it is based on a combination of Arp4 knockdowns in mammalian cells and nuclear transfer techniques in Xenopus oocyte nuclei. Readout for their experiments is the formation of nuclear actin bundles that can be visualized by a specific nuclear actin chromobody staining. In line with the notion that expression of Oct4 is dependent on nuclear actin polymerization, they show that changes in Arp4 expression directly affect expression of Oct4. In addition, they show preliminary data indicating that loss of Arp4 affects DNA damage since there is an increase in γH2AX positive foci. These results confirm the role of Arp4 as regulator of nuclear actin polymerization while correlating F-actin formation with possible yet unknown mechanisms of F-actin-dependent genome organization.
I read the paper with a lot of interest. It adds to previous work done by the authors. The conclusions are well supported by experimentation and not over-interpreted. In my opinion, apart from some minor language editing, this paper is suitable for publication in Cells.
Reviewer 2 Report
This manuscript concluded that Arp4 has a role as a suppressor of F-actin formation in the nucleus, thereby regulated gene expression via beta-catenin, SRF and DNA damage repair.
Overall, this manuscript provides interesting new information regarding the role of Arp4, but some more experiments are needed to strengthen the conclusion.
Figure 1a shows clearly more thick bundles of F-actin after Arp4 knockdown, but it is difficult to judge if the overall levels are also reduced. A western blot showing a difference in G-Actin vs F-actin in the nuclear fraction is needed to confirm this finding. The authors only used Hela cells in this study, a confirmation using at least one more mammalian cell line is needed, at least to reproduce figure 1a. In figure 5, the authors concluded that Arp4 has a role in DNA damage repair by only assessing a difference in gamma-H2AX levels at 1 h after Zeocin exposure. They quantified the intensity of gamma-H2AX as a ratio to DAPI. What is the rationale to normalize with DAPI but not intensity per nucleus? Is this a cell cycle dependent mechanism? There should be a discussion about this interpretation. Although statistically significant, the difference in figure5b is really minimal. It can also be interpreted that at 1 hour after the exposure there was just less formation of DSBs in Arp4KD cells. A conclusion about delayed DSB repair should be supported by a kinetics study. Eg. cells are exposed to Zeocin, then wash out followed by monitoring of DNA damage (gamma-H2AX ) over a time course eg. 2, 6, 8, 12 hours to monitor a difference in DNA repair. I suggest that counting number of cells with foci (fraction of cells with DSBs) or number of foci per cell can provide more useful information. The discussion part is insufficient. Please also provide more discussion whether dysregulation of Arp4 and Actin is related to any disease and how this finding will lead to further development in research such as in cancer research field etc.
Reviewer 3 Report
This study from Yamazaki et al. investigates the role of the Actin-related protein Arp4 in Nuclear F-actin functions, including regulation of gene expression and DSBs repair. The authors demonstrated that Arp4 is a negative regulator of Nuclear F-actin thorough a mechanism that does not interfere with nuclear actin levels. Furthermore, the authors showed that Arp4 inhibits F-actin-dependent reprogramming of mouse cells following transplantation into Xenopus oocytes, by inhibiting OCT4 transcription. In accordance, the Wnt/β-catenin pathway which is activated by nuclear F-actin was activated following Arp4 depletion.
Nuclear F-actin has emerged in the last few years as a critical regulator of many fundamental cellular processes including transcription, mitotic exit, DNA repair and genome stability (Andrin, Nucleus, 2012; Belin, eLife, 2015; Sun, PLoS One, 2017; Schrank, Nature, 2018; Caridi, Nature, 2018, Baarlink, Nat Cell Biol, 2017). Therefore, identifying Arp4 as a negative regulator of nuclear F-actin and characterizing its effect on reprogramming and DSB formation is of high importance.
Overall, this is an interesting paper, that adds to our understanding of nuclear F-actin regulation. However, a noted weakness that should be addressed is the lack of any data that sheds light on Arp4 regulation and its mode of action.
Major concerns:
The authors need to show whether the effect of Arp4 knockdown on nuclear F-actin is cell-cycle dependent. The authors should show by FACS whether/how Arp4 depletion affects the cell cycle. In addition, the authors should look whether inhibition of Arp4 promotes F-actin evenly thorough the cell cycle or is it specific to a certain phase (For example, by synchronizing the cells or co-express cell cycle marker like geminin or PCNA). The authors should shed some light on Arp4 regulation in response to reprogramming or DSB repair. To gain more physiological relevance the authors should show whether there is a drop in endogenous Arp4 level (protein and transcript) or change in its nuclear localization following DSB formation and during reprogramming.
Minor:
The authors should show the effect of Arp4 si-RNA on the protein level rather the transcription level. In Figure 1A, F-actin is demonstrated in the control cell. Indeed, actin filaments are thinner and less abundant in that example. However, the quantification represents a binary question (exist or not). The authors should explain what exactly they measured and provide the appropriate example. The authors should elaborate the material and methods section to a level that will allow the reader to understand how the experiments were done. For example, in Figure 1A were cells were co-transfected with si-RNA against Arp4 and with chromobody at the same time or was it done consecutively? How long after transfection cells were imaged? In Figure 4A, β-Catenin forms foci inside the nucleus. However, in Figure 4B the authors measured “relative amount” rather than “number or foci”. Why? This should be clarified.Author Response
Please see the attachment.

Round 2
Reviewer 2 Report
I would like to thank the authors for taking time to address most of the questions and performed some additional experiments. The manuscript has been significantly improved.
Regarding figure 5b, detecting the amount of gamma-H2AX at one time point still does not fully support the conclusion of DSB repair. The authors mentioned that they have done kinetic experiment and the result did not become more obvious. Thus, it is important to also include these results into figure 5 to support their conclusion.
Reviewer 3 Report
The paper can be accepted in the present form
Author Response
Reviewer 3
Comment #1: The paper can be accepted in the present form.
Response #1: We wish to express our appreciation for the review of our manuscript. We are glad that the manuscript has been significantly improved by the comments and suggestions by the Reviewer.